# Highly Sensitive and Specific Detection of Staphylococcal Enterotoxins SEA, SEG, SEH, and SEI by Immunoassay

**DOI:** 10.3390/toxins13020130

**Published:** 2021-02-09

**Authors:** Cécile Féraudet Tarisse, Céline Goulard-Huet, Yacine Nia, Karine Devilliers, Dominique Marcé, Chloé Dambrune, Donatien Lefebvre, Jacques-Antoine Hennekinne, Stéphanie Simon

**Affiliations:** 1Paris-Saclay University, CEA, INRAE, Medicines and Healthcare Technologies Department (DMTS), SPI, 91191 Gif-sur-Yvette, France; celine.goulard-huet@cea.fr (C.G.-H.); karine.devilliers@cea.fr (K.D.); dominique.marce@cea.fr (D.M.); chloe.g.d@hotmail.fr (C.D.); donatien.lefebvre@cea.fr (D.L.); stephanie.simon@cea.fr (S.S.); 2Laboratory for Food Safety, French Agency for Food, Environmental and Occupational Health & Safety (ANSES), Université Paris-Est, 94706 Maisons-Alfort, France; yacine.nia@anses.fr (Y.N.); jacques-antoine.hennekinne@anses.fr (J.-A.H.)

**Keywords:** staphylococcal enterotoxins, monoclonal antibody, enzyme immunoassay, lateral flow immunoassay

## Abstract

Staphylococcal food poisoning (SFP) is one of the most common foodborne diseases worldwide, resulting from the ingestion of staphylococcal enterotoxins (SEs), primarily SE type A (SEA), which is produced in food by enterotoxigenic strains of staphylococci, mainly *S. aureus*. Since newly identified SEs have been shown to have emetic properties and the genes encoding them have been found in food involved in poisoning outbreaks, it is necessary to have reliable tools to prove the presence of the toxins themselves, to clarify the role played by these non-classical SEs, and to precisely document SFP outbreaks. We have produced and characterized monoclonal antibodies directed specifically against SE type G, H or I (SEG, SEH or SEI respectively) or SEA. With these antibodies, we have developed, for each of these four targets, highly sensitive, specific, and reliable 3-h sandwich enzyme immunoassays that we evaluated for their suitability for SE detection in different matrices (bacterial cultures of *S. aureus*, contaminated food, human samples) for different purposes (strain characterization, food safety, biological threat detection, diagnosis). We also initiated and described for the first time the development of monoplex and quintuplex (SEA, SE type B (SEB), SEG, SEH, and SEI) lateral flow immunoassays for these new staphylococcal enterotoxins. The detection limits in buffer were under 10 pg/mL (0.4 pM) by enzyme immunoassays and at least 300 pg/mL (11 pM) by immunochromatography for all target toxins with no cross-reactivity observed. Spiking studies and/or bacterial supernatant analysis demonstrated the applicability of the developed methods, which could become reliable detection tools for the routine investigation of SEG, SEH, and SEI.

## 1. Introduction

Staphylococcal food poisoning (SFP) caused by ingestion of preformed staphylococcal enterotoxins (SEs) is among the most prevalent bacterial foodborne illnesses, with important public health, food safety, and economic impacts [1]. Symptoms are of rapid onset (30 min to 8 h after ingestion) and include violent vomiting, nausea, and abdominal cramping, with or without diarrhea. The illness is usually self-limiting and only occasionally needs hospitalization for very young, elderly, or immune-compromised patients.

*Staphylococcus* species are naturally present in the environment (air, dust, water) and belong to the normal flora of skin and mucous membrane of humans, mammals, and birds. As a consequence, SFP results either from a primary contamination of raw food (milk or meat) or from poor hygiene handling/inadequate storage of food (particularly mixed food, processed meats, and dairy products) contaminated with enterotoxigenic strains of coagulase-positive staphylococci (CPS), mainly *Staphylococcus aureus*. Generally, the production of SE(s) is observed when CPS enumeration reaches 10^5^ colony-forming units per gram (cfu/g) [2]. For this reason, criteria dedicated to the enumeration of CPS and the detection of SEs have been set down by manufacturers and competent authorities to define hygiene processes as well as food safety criteria. For example, EU Commission Regulation (EC) No. 2073/2005 amended by Commission Regulation (EC) No. 229/2019 relative to microbiological criteria for food [3] indicates, for milk and milk products, that detection of SEs must be performed when the CPS count exceeds 10^5^ cfu/g. The confirmed the detection of SEs in any foodstuff represents a potential hazard for human health as defined by Article 14 of Regulation (EC) No. 178/2002 [4]. However, low CPS levels cannot exclude the presence of preformed SE(s) in food, which can induce an SFP [5,6]: indeed, in addition to superantigenic activity, the extracellular SEs produced by enterotoxigenic strains of *S. aureus* are highly resistant to denaturation (freezing, drying, acid and heat treatments, digestive proteolysis) [2,5]. This property allows them (i) to remain intact in contaminated foods and trigger SFP when the SE-secreting bacteria themselves are destroyed [6] and (ii) to remain active in the digestive tract after ingestion [2]. Furthermore, the SEs can cause illness even at very low dose ranging from 20 to 100 ng (and down to 6.1 ng for SE type A (SEA) using modeling (Benchmark Dose)) [2,7]. In consequence, the conclusive confirmation of an SFP outbreak relies on the detection of SE toxin(s) in food remnants. Finally, notification of foodborne outbreaks is mandatory in many countries (e.g., for European Union Member States) [8].

To date, 27 distinct enterotoxins have been described as globular single-chain proteins from 22 to 29 kDa and named sequentially in the order of their discovery (except for SE type F (SEF), lacking emetic activity, which was later renamed TSST1). For 19 of them, an emetic activity was demonstrated (SE type A to E (SEA to SEE), and more recently SE type G (SEG), H (SEH), I (SEI), K to T (SEK to SET), and Y (SEY)) [9,10,11,12,13,14], whereas others have not yet been confirmed to cause emesis and are thus more appropriately referred to as SE-like toxins (SE*l*J, SE*l*U, SE*l*V, SE*l*X, SE*l*Z, SE*l*26 and SE*l*27) [15,16,17,18]. SEA, alone or in association with other SE(s), is the most frequently involved (>75%) in SFP outbreaks worldwide. SEB is not only involved in food poisoning, but it was also historically identified as a potential biological weapon for bioterrorism (classified as category B by the Centers for Disease Control and Prevention). However, from a clinical point of view, all SE toxins exhibit the same incapacitating properties as SEB. Thus, there is a need for rapid, sensitive, and specific methods for the detection of ideally all toxin types for dual applications (natural contamination or intended misuse).

Four main types of methods are used to detect bacterial SE toxins and characterize SFP outbreaks: bioassays, molecular biology, mass spectrometry, and immunological techniques [2,19]. Bioassays in animal and/or cell cultures, despite being among the earliest methods developed to detect SEs, are no longer appropriate for confirmation of an SFP outbreak because of insufficient sensitivity, absence of specificity, poor practicability, and, for animal bioassays, not being ethically acceptable. Biotyping methods (such as PCR and Whole Genome Sequencing), although reliable, have two major limitations: (i) they first require an isolation of staphylococcal strains from food, and (ii) they are based on DNA analysis but do not provide any information on the expression of SEs themselves in the ingested food. Mass spectrometry-based methods have newly emerged as a promising technique, but they still suffer from interference in complex matrices (requiring extensive sample preparation), high cost analysis, low throughput, and the need for highly qualified personnel. The last and most commonly used method for detecting SEs in food is based on the use of polyclonal or monoclonal antibodies specific for these toxins (through immunoassays, latex agglutination, immunoblotting, or immunochromatography). Commercial kits have been available for more than 15 years to detect the five so-called “classical SEs”, i.e., SEA, SEB, SEC, SED, and/or SEE, based on either enzyme immunoassays (using chromogenic substrates (ELISA) or fluorescent dyes (ELFA) [20,21]) or reverse passive latex agglutination (RPLA [22]). Since 2017, an international standard (EN ISO 19020) [23] has been applied to the global qualitative detection of the five classical SEs in food matrices using two commercially available immunoassay tests, Vidas SET2 and Ridascreen SET Total. However, no information on the type of enterotoxin detected can be obtained. These are the official screening and confirmatory methods for SE detection in food and are routinely used in food safety laboratories to identify and confirm SFP outbreaks [8].

However, recently and throughout the world, several *S. aureus* isolates derived from food poisoning were shown to be negative for SEA to SEE toxins, but, thanks to extensive genetic analysis, they were shown to harbor “non-classical SE” genes, indicating that the recently described SEs and SE*l*s could also be the causative agents of SFP outbreaks. Several scientific reports describe these atypical strains associated with food poisoning and carrying non-classical enterotoxin genes: *seh* [24], one or more of the *seg*, *seh*, or *sei* genes [25,26,27], *selj*, *ser*, *ses,* and *set* [11] and, more recently, the enterotoxin gene cluster (*egc*) including a group of superantigens (*seg*, *sei*, *sem*, *sen*, *seo* in addition or not to *sel* and *selu* genes) assembled into operon-like clusters [28,29,30]. To clarify the role played by these newly identified SE/SE*l* toxins in SFP, the development of reliable non-classical enterotoxin detection methods is essential. That is why a few in-house immunoanalytical methods have been developed and described in the literature to detect one or more of these toxins. They are mainly represented by sandwich ELISA using rabbit polyclonal antibodies, which are reported to detect SEH [31,32], SEG, SEH, and SEI [33] or SE*l*J [34]. The use of such polyclonal antibodies can be considered as a disadvantage for ELISA robustness due to the increased probability of cross-reaction with other SE(s) and even other unrelated proteins (false positives) and higher risk of batch-to-batch variability compared to monoclonal antibodies. As a first improvement, polyclonal chicken immunoglobulins (IgY) were used for SEG detection to eliminate cross-interference induced by staphylococcal protein A with mammalian antibodies [35]. Lately, to increase detection specificity, at least one monoclonal murine antibody was inserted in the newly reported developments of sandwich ELISA described for the detection of SEH [36] or SEI [37]. Finally, a multiplex prototype made of hydrogel immunobiochips has also been designed and reported to detect simultaneously seven enterotoxins, including SEG and SEI [38], with a demonstrated feasibility. Thanks to these in-house sandwich immunoassays, *S. aureus* isolates involved in SFP were shown to be able to produce in vitro these “new enterotoxins” such as SEH [31,32,33], low levels of SEI [33,37] and SEG [33,35], or even SE*l*J [34]. The involvement of SEH in food poisoning outbreaks has since been well established, as this toxin was directly detected in food products responsible for two SFP. The first one took place in Norway, where an estimated 55 ng of SEH/g of mashed potato made with raw bovine milk led to the poisoning of eight individuals [39]. In the second, large outbreak (poisoning of more than 10,000 individuals) in Japan, a combination of small amounts of SEH and SEA was suspected: the quantified 80 ng of SEA was considered insufficient to cause this massive outbreak, except if we take into account the nearly equal amounts of SEH concomitantly found in food remnants [40].

Given that SEG, SEH, and SEI have all been shown to induce emesis and to be involved in food poisoning, a qualitative commercial kit for the combined and undifferentiated detection of these three toxins has recently been developed, and its performances were evaluated: the VIDAS^®^ Staph Enterotoxin III (SET3) (bioMérieux, Craponne, France) [41]. However, this grouped and undifferentiated detection does not fully meet the needs for an individual detection of SE(s) for precise poisoning toxin identification required for a deeper epidemiological analysis.

We sought to develop suitable methods for the sensitive, specific and, above all, individual detection of each SEG, SEH, and SEI enterotoxin, using sandwich enzyme immunoassays and immunochromatographic tests, to enable future studies to determine the individual role of these toxins in SFP outbreaks. This need was clearly expressed by the French National Reference Laboratory (NRL) and European Reference Laboratory (EURL) for Coagulase Positive Staphylococci. Since SEA is involved in >75% of SFP, it was considered relevant to combine SEG, SEH, and SEI detection with the one for SEA. We took advantage of producing our own monoclonal antibodies to follow a custom designed hybridoma triage process enabling the selection of mAbs with the desired specificity.

The enzymatic immunoassays developed in this study with these new monoclonal antibodies were evaluated for their suitability for the detection of SEA, SEG, SEH, and SEI in different matrices for different purposes: (i) bacterial cultures from *S. aureus* (reference strains or isolates from food poisoning outbreaks) for the determination of specificity and strain characterization, (ii) certified reference materials [42,43,44] and spiked foodstuffs for food safety and biological threat detection, and (iii) human samples for diagnosis. Concerning this last point, to date, there is no diagnostic test kit available for clinical samples such as diarrhea or vomit. The only reported study [45] shows that the two evaluated commercial tests cannot be adapted from their primary use (i.e., enterotoxin detection in food matrices), as they were not able to detect SEs directly in stool but only after *S. aureus* strain isolation. The enzymatic immunoassays developed in this study were evaluated for their usefulness as a diagnostic tool.

Finally, we initiated the development of monoplex and multiplex lateral flow immunoassays for the detection of these new staphylococcal enterotoxins. To our knowledge, to date, only immunochromatographic tests for the detection of “classical SEs” have been described [46,47], but they are none for SEG, SEH, and SEI enterotoxins. Such rapid tests present advantages that could be exploited in a future study: robustness, easy transfer into a commercialized version, stability for more than 2 years without refrigeration, user-friendliness, high performance, low cost, rapidity (results in less than 30 min), and no requirements for supporting technical infrastructure.

## 2. Results

### 2.1. Production and Characterization of 79 Monoclonal Antibodies Directed against SEA, SEG, SEH, or SEI Toxins

Monoclonal antibodies (mAbs) were raised in *Biozzi* mice by immunization with recombinant staphylococcal enterotoxins SEA, SEG, SEH, or SEI that were expressed in *E. coli*, purified thanks to their poly-histidine tag by nickel affinity chromatography (gel electrophoresis shown in Appendix A), and fully characterized by bottom–up mass spectrometry (D. Lefebvre et al., in revision, Journal of Agricultural and Food Chemistry). The 79 mAbs produced were named in relation to their target toxin followed by a number, and they include 11 mAbs directed against SEA, 20 against SEG, 18 against SEH, and 30 against SEI (Appendix A). Purified mAbs were characterized by sandwich immunoassays (combinatorial analysis of all possible pairs of mAbs), Western blotting, epitope mapping, and affinity determination by bio-layer interferometry.

Thanks to a double screening immunotest systematically performed during the hybridoma selection and cloning process (see Methods and Appendix A), each mAb was selected for its specificity for its target toxin. It was confirmed in the complementary binding study for all combinations of mAbs where no cross-reactivity toward untargeted recombinant toxins was observed (data not shown). Thereby, six sandwich immunoassays were identified as specific for the detection of lab-made recombinant SEA and commercial SEA, without cross-reactivity toward recombinant SEG, SEH, or SEI. More than 100 sandwich immunoassays allowed the specific and sensitive detection (<0.4 ng/mL) of SEG, SEH, and SEI without cross-reactivities with each other or with lab-made recombinant and commercial SEA. With these complementary binding studies, we identified groups of mAbs possibly targeting similar, overlapping, or nearby epitopes of the specific toxin (Appendix A), as they could not bind simultaneously to their target and detect the toxin only when used in combination with other groups of mAbs. However, no linear epitope could be identified by epitope mapping (Pepscan) for the most used mAbs (i.e., involved in Figure 1) (data not shown) despite good performance in Western blot experiments (data not shown).

Kinetic parameters of the most used mAbs (combinatorial analysis criteria) were determined by bio-layer interferometry in multi-cycle kinetics using lab-made recombinant target toxins and commercial SEA toxin as antigen (Appendix A). The equilibrium dissociation constant K_D_ was calculated from the ratio of dissociation/association kinetics (k_off_/k_on_). Among the four best anti-SEA mAbs identified in the sandwich combinatorial immunoanalysis, two-SEA7 and SEA12-stood out, with notable and quite similar affinity constants with K_D_ near 5 × 10^−11^ M and 1 × 10^−11^ M, respectively, toward lab-made recombinant SEA and commercial SEA. Nevertheless, we could observe a slight difference in the K_D_ of each mAb toward the commercial versus recombinant form of SEA. This could be explained by the different origins of the toxin, one being natural and commercially highly purified, the other being recombinant with a 3D conformation slightly different from the natural one, and lastly by the precision of concentration measurement (and the molar extinction coefficient used). Such a phenomenon is commonly observed [48]. Concerning the best antibodies directed against SEG, they exhibited similar K_D_ values in the range of 10^−10^ M, except for SEG21, which showed a slightly less notable K_D_ of 3.9 × 10^−9^ M. Another exception was SEG27, with a noteworthy K_D_ of 4.8 × 10^−11^ M, which is associated with a combined low dissociation rate of 8.7 × 10^−5^ s^−1^ and fast association of 1.8 × 10^6^ M^−1^.s^−1^. The K_D_ of the best anti-SEH mAbs ranged around 10^−10^ M, except for SEH1 and SEH4, which had one log lower equilibrium dissociation constants (approximately 10^−9^ M) as opposed to SEH11 and SEH14, which exhibited slightly better K_D_ values around 6 × 10^−11^ M with a fast association rate (approximately 2 × 10^6^ M^−1^.s^−1^). Lastly, the best anti-SEI mAbs showed equilibrium dissociation constants around 10^−10^ M, except for SEI26 and SEI32, which had lower K_D_ values around 10^−9^ M.

### 2.2. Development of Four Sandwich Enzyme Immunoassays for the Specific Individual Detection of SEA, SEG, SEH, and SEI Toxins

#### 2.2.1. Selection of Best mAb Pairs

To select and develop optimal individual toxin immunoassay detection tests, the best pairs of mAbs identified in the combinatorial analysis (see Methods) were further evaluated for sensitivity, specificity, and signal-to-noise ratio performances in a 3-h format for the detection of both recombinant toxin and its native form from *S. aureus* supernatants.

The 3-h sequential format was decided upon to meet the requirements of matrix interference problems (in both food safety and human diagnosis purposes), relative ease of use, and ease of transfer to other labs. This format consisted of a first 1-h incubation of samples on the specific antibody-coated plates, followed, after washes, by a 1-h reaction with biotinylated anti-toxin mAb and a 30-min reaction with enzyme-labeled streptavidin. The last 30 min comprised colorimetric reaction and reading.

Dilution series of target and non-specific toxins from both recombinant sources and *S. aureus* supernatants (Table 1) were tested in this format to select the best capture and detection mAb pair for each SEA, SEG, SEH, and SEI toxin.

Concerning SEA detection, two mAb pairs stood out among all the combinations, involving SEA7 as capture antibody, with biotinylated SEA5 or SEA12 as tracer. The selected mAb pair (capture SEA7/biotinylated SEA5) showed better sensitivities in comparison to the second mAb pair for the detection of both recombinant and commercial SEA (Table 2), and most importantly, a three-fold better signal-to-noise ratio for the detection of native SEA from FRI S6 *S. aureus* strain supernatant without significant cross-reactivities with the other strains (Figure 1a).

Regarding SEG detection, eight mAb pairs were first selected from the combinatorial analysis and compared in the 3-h sequential format (Figure 1b), including SEG27, which showed a notable K_D_ (Appendix A) associated with good tracer performances (Appendix A). The best pair, which was composed of capture SEG41 and biotinylated SEG27, was selected as it exhibited the best signal-to-noise ratio for the detection of native SEG from the three *S. aureus* strains expressing this target toxin with high specificity (no cross-reactivity with the other strains, Figure 1b) and sensitivity performances (satisfying theoretical limit of detection (LoD) and limit of quantification (LoQ) for recombinant SEG, Table 2).

For SEH detection, seven mAb pairs were first selected from the combinatorial analysis including SEH11, SEH14 (antibodies with the best affinities, Appendix A), SEH1, and/or SEH19 (the best capture or tracer mAb respectively in the combinatorial analysis, Appendix A). The selected mAb pair (capture SEH1/biotinylated SEH19) exhibited a sensitivity among the best for the detection of lab-made recombinant SEH toxin with a theoretical LoD of approximately 2 pg/mL and a theoretical LoQ near 11 pg/mL (Table 2). Above all, it was the sandwich mAb pair that had the best signal-to-noise ratio for the detection of native SEH from the three available *S. aureus* strains expressing this target toxin, without cross-reactivity toward the other strains (Figure 1c).

Finally, concerning the detection of SEI, only eight mAb pairs among the best 66 enabling recombinant SEI overnight detection efficiently detected native SEI from *S. aureus* supernatants in a 3-h sequential format (despite detection of recombinant SEI). Among these eight sandwich tests, only two recognized the three *S. aureus* strains expressing native SEI (Figure 1d) with similar performances (Table 2): SEI27 as capture antibody combined with either biotinylated SEI26 or SEI32. The first mentioned mAb pair was selected because of slightly higher absorbance signals (data not shown).

#### 2.2.2. Performances of the Immunoassays

Performances (sensitivity, specificity, repeatability, in-house reproducibility, accuracy, and robustness) of the best-selected sandwich immunoassays for each of the four SE toxins were evaluated using the 3-h sequential format with poly-horseradish peroxidase-labeled streptavidin detection. The latter enzyme was chosen as a label because of its ease of transfer to other labs.

Sensitivities reached in dilution buffer (enzyme immunoassay buffer (EIA buffer)) are reported in Table 3 and illustrated in Figure 2. Theoretical limits of detection (LoD) and quantification (LoQ) of lab-made recombinant and commercial toxins were approximately 10-fold better for SEG and SEH (LoD of 0.21 pg/mL and 0.59 pg/mL, and LoQ of 0.66 pg/mL to 1.73 pg/mL respectively) than for SEA and SEI (LoD from 2.0 pg/mL to 5.4 pg/mL and LoQ from 6.0 pg/mL to 17.3 pg/mL) in EIA buffer. The difference in sensitivities presented in Table 2 and Table 3 can be explained both by the use of different dilution medium/buffer and different enzyme label (acetylcholinesterase (AChE) versus poly-horseradish peroxidase (HRP)). For lab-made recombinant toxins, we also evaluated in ≥17 independent experiments the experimental LoD and LoQ values (defined as the experimental lowest concentrations giving, in almost ≥95% of cases, a signal greater than the mean of nonspecific binding + 3 or 10 standard deviations for LoD or LoQ, respectively). They are approximately two-fold higher than theoretical values (Table 3), such difference being common and expected [50].

Repeatability, in-house reproducibility (assays performed the same day or on different days, respectively) and accuracy (bias) were evaluated by measuring, using standard titration curves, six times and on five different days, different concentrations of commercial SEA and lab-made recombinant SEA, SEG, SEH, and SEI toxins in EIA buffer from experimental 1 × LoQ or 2 × LoQ to the beginning of saturation. These tests showed good accuracy (from 87.1% to 101.3%, acceptable range being between 80% and 120% [51]). They also presented satisfactory precision in the working concentration range of each toxin with coefficients of variation <16% at low concentrations and <10% at higher concentrations (Table 4).

The specificity of the immunoassays was evaluated through the quantitative measurements of SE toxins in *S. aureus* supernatants from eight different strains cultivated in BHI medium for 16 h. The results, presented in Table 5, showed no nonspecific cross-reactivity and enabled detection of the expected toxins (Table 1).

#### 2.2.3. Robustness in Complex Matrices

To validate the use of the immunoassays for food safety, biological threat detection, and human diagnosis purposes, artificially contaminated (spiked) samples of various origins were tested. Dairy product (semi-skimmed milk) and sugary drink (apple juice) were used as representative of liquid matrices. Dairy food (“Morbier” cheese), starch groceries (chili with rice without meat), and meat (ham and chili with beef without rice) were selected as solid foodstuffs. Finally, human samples consisted of diarrhea, stool, and vomit that were prepared, as the solid food products, by grinding to yield a 10% (*W*/*V*) buffered homogenate and then clarified by centrifugation. All these matrices were analyzed before and after spiking with each of the recombinant lab-made SE toxins with the 3-h sequential immunoassays. Figure 3 illustrates the performance heterogeneity for SE toxin detection in spiked matrices and the importance of minimal dilution to reduce the matrix effect. Semi-skimmed milk and vomit samples were the least interfering matrices, with no dilution to be applied to reach a signal and sensitivity similar to those reached in EIA buffer (with the exception of SEI immunoassay, for which a 2-fold dilution of milk seemed necessary). In contrast, diarrhea and human stool were systematically interfering for all four immunoassays, inducing both specific signal reduction (Figure 3) and higher non-specific binding signals (particularly for SEG ELISA, data not shown). A 20-fold dilution of diarrhea in EIA buffer seemed to be the minimum to recover 80% of expected signals (but still insufficient for stool samples). Between these two extremes, performance heterogeneity in SE toxin detection was observed. Apple juice was interfering for SEA, SEI, and to a lesser extent SEH immunoassays, and this matrix interference was rapidly overcome by the dilution effect: 2- to 5-fold dilution in EIA buffer restored the sensitivity reached in buffer. Morbier cheese, ham, and starchy food affected SEA and SEH detection, whereas starchy food impacted very slightly the SEI immunoassay, and once again, a 2- to 5-fold dilution of samples in EIA buffer could easily impair all these matrix interferences.

The enzyme immunoassays were transferred to the French NRL for CPS and were challenged by the NRL on three certified reference materials (CRM) (IRMM-359a, IRMM-359b, and IRMM-359c) and on three other cheese samples from the NRL collection (two freeze-dried Emmental cheese powders and one naturally contaminated Morbier cheese). Thus, data obtained with the ELISA tests were compared to data obtained by the EN ISO 19020 using the same samples.

For blank CRM (IRMM-359a), staphylococcal enterotoxins were not detected whatever the method implemented, confirming the specificity of the ELISA method against the international standard. For CRM IRMM-359b and IRMM-359c contaminated with SEA, the results obtained by the international standard showed a positive response but without identification of the type of enterotoxin present. This qualitative result is satisfactory (according to the producer’s certificate). Analyses performed using the immunoassays in our laboratory and at NRL for CPS showed the detection of SEA in both IRMM-359b and IRMM-359c, and no detection of enterotoxins SEG, SEH, and SEI, thus confirming the specificity of the ELISAs (Table 6). Quantitatively, SEA concentrations measured in CRMs obtained in both laboratories were highly comparable and in good agreement with the values indicated in the producer’s certificates.

For Emmental cheeses spiked with SEA, the EN ISO 19020 standard gave a positive response, thus indicating the presence of enterotoxins SEA to SEE without specifying the type of enterotoxin present. The immunoassays confirmed the presence of SEA in these two Emmental cheeses, and SEG, SEH, and SEI enterotoxins were not detected (Table 6). For Morbier cheese naturally contaminated by SED, the EN ISO 19020 standard gave a positive response indicating the presence of enterotoxins SEA to SEE without specifying the type of enterotoxin present. The ELISAs showed that enterotoxins SEA, SEG, SEH, and SEI were not detected, highlighting the high specificity of the four immunoassays, as only SED is present in the Morbier cheese (Table 6).

### 2.3. Development of Immunochromatographic Tests for the Monoplex and Multiplex Detection of SEA, SEB, SEG, SEH, and SEI Toxins

The best mAb combinations evaluated for the 3-h sequential immunoassays (Figure 1) were compared in an immunochromatographic format for the individual detection of each of the four target lab-made recombinant SEs (data not shown). The best sensitivities in buffer without any background noise were respectively obtained for the following capture/colloidal gold conjugate mAb pairs: SEA7/SEA12, SEG41/SEG27, SEH14/SEH19, and SEI44/SEI32 (Figure 4). Except for SEG detection, the mAb pairs selected for immunochromatographic purposes were different from those used for ELISA and were consistent with their kinetic parameters (Appendix A). Indeed, due to their short duration and extremely fast target/antibody interaction time over the migration process, lateral flow tests require antibodies with high association rate constants k_on_. This explanation is in agreement with SEA7/SEA12 mAb pair selection involving the two compatible anti-SEA antibodies showing the highest k_on_. Concerning SEG detection, the selected mAb pair was unchanged, as it already comprised the two antibodies with the best association rate constants. Antibody SEH14 was also the one with the higher k_on_, and it was associated with the first-rate and unequalled tracer SEH19. Finally, as SEI39 mAb induced a nonspecific signal (data not shown), capture SEI44 (which gave results similar to SEI36) was selected in combination with SEI32.

An LoD close to 0.3, 0.01, 0.02, and 0.1 ng/mL was reached in buffer in 30 min respectively for SEA, SEG, SEH, and SEI toxins (Figure 4), which was approximately 1 or 2 logs less sensitive than the 3-h sequential immunoenzymatic test. Specificity was satisfying, as no cross-reactivities were identified with the available recombinant and commercial toxins (Figure 4). Performances of these monoplex immunochromatographic tests were also evaluated with the eight culture supernatants of *S. aureus* and gave the expected results (Figure 4). Results obtained in Figure 4 by lateral flow immunoassay and in Figure 1 by ELISA cannot be compared directly, as the *S. aureus* culture conditions and sample buffer composition (detergent presence or not) were different.

The monoplex immunochromatographic tests were further evaluated in more complex matrices (Figure 5) showing no or slight (semi-skimmed milk), medium (apple juice), or important (diarrhea) matrix effect in previous ELISA experiments (Figure 3). For matrices with small fragments (apple juice sediment, raw diarrhea), a nonspecific band appeared in the lower part of the membrane (i) due to retention of the colloidal gold conjugate in the sample pad because of a physical barrier made of matrix fragments (too large to migrate along the membrane) and (ii) resulting in lower intensities for both test and control lines. No false positive was observed excepted for milk using the SEG immunochromatographic test (dilution of milk could overcome this false positive, data not shown). In contrast to ELISA (Figure 3), there was no marked performance heterogeneity in SE toxin detection between matrices and toxin types, but a relative homogeneous approximately 3-fold decrease in sensitivity in comparison with performance in buffer (and up to a 10-fold decrease in spiked raw diarrhea, Figure 5), highlighting the robustness of lateral flow immunoassays.

To go further in the development of immunochromatographic assays, multiplex tests were developed for the simultaneous detection of SEA, SEB, SEG, SEH, and SEI, using the four above-mentioned selected mAb pairs with the addition of the previously identified SEB27/SEB26 mAb pair directed against staphylococcal enterotoxin B [47]. A pre-industrial batch (NG-Biotech, France) of these multiplex strips included in cassettes was produced and tested (Figure 6).

Limits of detection close to 0.3–0.9, 0.1, 0.1, 0.3, and 0.3 ng/mL were reached for SEA, SEB, SEG, SEH, and SEI enterotoxins respectively in buffer in 30 min (Figure 6). They were up to one log less sensitive than lab-made monoplex strips, and this difference could be explained by the horizontal migration in the cassette (versus vertical migration for lab-made strips) and lyophilized tracer resuspended by the sample flow (versus 5-min incubation between sample and tracer before performing the lab-made monoplex lateral flow test), which both reduced target/antibody interaction duration.

Further evaluation is needed to complete the evaluation of this multiplex immunochromatographic test using a large panel of *S. aureus* strain supernatants, complex matrices (food), and real naturally contaminated samples. These studies are currently underway and will be described in a further publication.

## 3. Discussion

Reliable methods for the sensitive, specific, but also individual detection of new enterotoxins are needed to document precisely SFP outbreaks and determine the individual role of these new enterotoxins or SE-like toxins so as to achieve more in-depth epidemiological analysis. In the present study, we produced specific monoclonal antibodies against the new enterotoxins SEG, SEH, and SEI and also classical SEA (responsible for >75% SFP outbreaks worldwide) to develop four 3-h format sandwich ELISAs that detect each of these toxins individually, as well as monoplex and quintuplex (SEA, SEB, SEG, SEH, and SEI) lateral flow immunoassays performed in 30 min.

We paid particular attention to the specificity of the mAbs produced as these four enterotoxins share 27 to 39% amino acid sequence identity [5]. Throughout hybridoma selection and cloning of monoclonal antibodies, a double screening immunotest was systematically performed until the selection of the 79 mAbs using both lab-made *E. coli* recombinant nonspecific and target toxins to ensure specificity for the target SE. As we had previously observed that the best mAb pair for the detection of a recombinant protein is not systematically the best for the detection of the natural protein, the final selection of antibody pairs was made by using native SEs from eight different *S. aureus* strain supernatants. This step was crucial and directly drove the choice of the final selected mAb pair, which otherwise might have been different and possibly less effective, particularly for SEI sandwich ELISA where only eight mAb pairs out of the 66 selected recognized native SEI from at least one of the three *S. aureus* strains harboring the *sei* gene and only two antibody pairs recognized these three strains. Indeed, SEI variants, as well as variants of other SE types, are known [52], and the production of mAbs that recognize all variants of a single SE type is therefore challenging. To challenge our developed immunoassays with this SE variant diversity and to check wisely the immunotest specificity toward the 27 existing SEs, further experiments are ongoing at the French National Reference Laboratory for Coagulase Positive Staphylococci for in-depth immunoassay validation. These tests are performed using a large *S. aureus* cell bank producing different SEs (cross-reactivity validation and variant inclusivity performance) to evaluate the specificity of the immunoassays toward SEs showing high amino acid sequence identity (example: 50% or 83% identity between SEA and SED, or SEA and SEE, respectively [5]). Great efforts are also made to fine-tune matrix interference study for precise robustness evaluation in real or spiked food extracts obtained according to Standard NF EN ISO 19020 [53]. These results will be presented in a future publication.

In our easily transferable 3-h sequential enzyme immunoassays, experimental detection limits in buffer reached 4.1 and 12.3 pg/mL for lab-made recombinant and commercial SEA respectively, and 0.41, 1.37, and 4.1 pg/mL for lab-made recombinant SEG, SEH, and SEI, respectively. Experimental quantification limits for these same targets were 12.3 and 37.0 pg/mL for the two SEA, respectively, and 1.23, 4.12, and 37.0 pg/mL for SEG, SEH, and SEI, respectively. In the range from approximately 2×LoQ_experimental_ to the beginning of binding saturation, intra-assay coefficients of variation (CV) were below 10% (repeatability) and inter-assay CV was below 15% (reproducibility), fulfilling requirements for a reliable test. Sensitivities reached in buffer by the enzyme immunoassays are 50- (for SEI) to 500-fold (for SEG) better than the best ones described in the literature. Indeed, Hait et al. [41] described an LoD of 0.2 ng/mL for SEG in both buffer and milk using the new VIDAS^®^ Staph Enterotoxin III (SET3) kit (bioMérieux, Craponne, France), whereas 0.5 ng/mL [33] and 1 ng/mL [35] were reported for lab-made SEG sandwich ELISAs using rabbit polyclonal antibodies and chicken anti-SEG IgY, respectively. SEH detection sensitivities have improved over time from 2.5 ng/mL [31] to 1 ng/mL [32,33] and recently reached 0.4 ng/mL [41] and 0.2 ng/mL [36] in buffer using in-house immunoassays. Finally, the best-described SEI detection sensitivity in buffer was also 0.2 ng/mL using, as for SEG, the VIDAS^®^ Staph Enterotoxin III (SET3) kit [41], followed by 0.5 ng/mL [37] and 1 ng/mL [33] using in-house immunoassays.

The complexity of some food and human sample matrices, along with other factors, ineluctably influences the limits of detection of sandwich immunoassays. For instance, Hait described a 1-log reduction in sensitivity in the six spiked and extracted chosen foodstuffs, and a detection level near 2 ng/mL for SEG, SEH, and SEI using the SET3 kit, with only one substandard exception [41]. In our series of experiments, we did not perform any SE extraction from these complex matrices but only ground and clarified them to produce 10% homogenates. We also observed substantial interference from some food matrices for certain SEs (none for SEG), but that could be easily overcome by applying a 2- to 5-fold dilution in buffer. The most critical matrix was human feces: despite possible but considerably less sensitive detection, diarrhea and stool were clearly outliers as previously identified [45]. To overcome this problem, an adapted sample preparation (to be defined) done before the test could provide a solution. For these particularly complex matrices, another alternative relies on the use of immunochromatographic tests known to be more robust than ELISA and thus possibly offering a better analytical performance, especially if integrated within a next-generation lateral flow device [54]. To date and to our knowledge, no quantitative information is available concerning SE contamination levels possibly found in clinical samples such as vomit or diarrhea from poisoned patients. Thus, it is difficult to know if our immunotest could be used for this diagnostic purpose.

To our knowledge, to date, only immunochromatographic tests for the detection of “classical SEs” have been described [46,47], but there have been none for SEG, SEH, and SEI enterotoxins at a single or multiplex detection level. The sensitivities of the monoplex lateral flow immunoassays were 0.3 ng/mL for SEA, 0.01 ng/mL for SEG, 0.02 ng/mL for SEH, and 0.1 ng/mL for SEI enterotoxins in buffer in 30 min. In more complex matrices, sensitivities were decreased approximately 3-fold and up to 10-fold in artificially contaminated raw diarrhea. In the multiplex pre-industrial cassette format, due to the more stringent conditions for horizontal lateral flow assay and manufactured format (i.e., with lyophilized colloidal gold conjugate mAb), these sensitivities were slightly reduced to 0.9 ng/mL for SEA, 0.1 ng/mL for SEG, 0.3 ng/mL for SEH, and 0.3 ng/mL for SEI, with the addition of SEB detection with a sensitivity limit of 0.1 ng/mL. Such rapid tests open new perspectives for applications where user friendliness, low cost, and faster detection are required, with little supporting infrastructure, or in decentralized testing environments with advanced facilitative technologies [54].

## 4. Conclusions

The immunoassays we have developed using monoclonal antibodies could become reliable detection tools for the routine investigation of SEA, SEG, SEH, and SEI in confirmation of staphylococcal food poisoning, for national reference laboratories and academic research, and possibly for diagnosis and biological threat detection purposes.

## 5. Materials and Methods

### 5.1. Bacterial Cultures

The CPS strains used in this study are listed in Table 1. They were grown in Luria–Bertani (LB) or Brain Heart Infusion (BHI) medium at 37 °C for 16 h with shaking. Cells were separated from the medium by centrifugation at 5500 rpm for 15 min. The supernatants were sterilized by filtration through a 0.2-μm syringe filter, and interfering protein A naturally produced by *S. aureus* was neutralized by overnight incubation at 4 °C with 5% (*V*/*V*) Ehrlich ascites fluid or normal rabbit serum (as previously described [37,55]) before immunoanalysis testing.

### 5.2. Recombinant His-Tagged SEA, SEG, SEH, and SEI Production and Purification

The DNA sequences of the *sea* (GenBank M18970.1), *seg* (GenBank CP001781.1), *seh* (GenBank AY345144.1), and *sei* (GenBank CP001781.1) genes from *S. aureus* (excluding signal sequence) were codon optimized for *E. coli* protein expression. NdeI and XhoI restriction sites were respectively added at the 5′ end and 3′ end of the sequence. The resulting *sea*, *seg*, *seh,* and *sei* gene sequences were synthesized and inserted into pUC57 plasmid (Genecust, Boynes, France). The four genes (sequentially digested with NdeI and then XhoI) were purified and ligated separately into the isopropyl-β-thiogalactoside (IPTG) inducible pET22b(+) vector (Novagen), allowing the insertion of a poly-histidine tag sequence at the 3′ end of the genes. The pET22b(+)-*sea*, -*seg*, -*seh*, and -*sei* recombinant plasmids were used separately to transform competent *E. coli* DH5α cells (from Invitrogen, Thermo Fisher Scientific, Illkirch, France) using a standard heat-shock method. Recombinant clones were screened by PCR for the presence of toxin genes and the sequences of the plasmids pET22b(+)-*sea*,- *seg*, -*seh* and -*sei* were confirmed by DNA sequencing (Eurofins Genomics, Les Ulis, France). Then, *E. coli* BL21 (DE3) pLysS cells (from Invitrogen, Thermo Fisher Scientific, Illkirch, France) were transformed with confirmed recombinants. Then, conditions for optimal toxin expression were determined for the best resulting transformants in Luria–Bertani Broth (LB) with 50 μg/mL ampicillin and with isopropyl-β-thiogalactoside (IPTG) induction when optical density at 600 nm reached 0.6.

For the first experiments using recombinant SEA, i.e., for mouse immunization with SEA and for anti-SEA hybridoma screening, recombinant SEA was purified from inclusion bodies, as it is a production strategy with high yield. However, for all other following steps, i.e., for anti-SEA antibody characterization and SEA immunoassay development, but also for all work using other target SEs, recombinant enterotoxins (SEA, SEG, SEH, and SEI) were purified from *E. coli* cytosol. Indeed, despite weaker production yields, refolding of cytosolic proteins is more adequate than the one from inclusion bodies made of densely packed aggregated proteins that need to be denatured in urea for purification.

Optimal SEA production in *E. coli* inclusion bodies was reached at 37 °C after induction with 50 μM IPTG for 4 h. The pellet (from 300 mL culture) was resuspended in 30 mL of solubilization buffer (50 mM Tris-HCl pH 8, 500 mM NaCl, 8 M urea and 1 mM protease inhibitor (4-(2-aminoethyl)-benzenesulfonyl fluoride (AEBSF), Interchim, Montluçon, France)) and, after two 15-sec pulses of sonication, it was allowed to dissolve for 1 h at 37 °C with shaking. After centrifugation at 20,000× *g* for 15 min at 4 °C, the supernatant was collected. Imidazole (20 mM final concentration) was added to the supernatant before loading on a previously equilibrated 1 mL nickel-charged nitrilotriacetic acid (Ni-NTA) chelate immobilized on agarose resin (Chelating Sepharose FastFlow charged with Ni^2+^, GE Healthcare, Buc, France).

The larger quantities of SEA, SEG, SEH, and SEI in the cytosol were optimally produced at 30 °C after induction with 100 μM IPTG (for SEG, SEH, and SEI) and 500 μM IPTG (for SEA) for 4 h (for SEA, SEG, and SEH) and overnight (for SEI). All the cultures were pelleted by centrifugation at 2500× *g* for 20 min at 4 °C. Bacterial pellets (from 300 mL culture) were resuspended in 30 mL of 50 mM Tris-HCl pH 8, 100 mM NaCl containing 1 mM of protease inhibitor (AEBSF, Interchim, Montluçon, France). Then, the bacterial suspensions were sonicated (2 pulses of 15 sec) and centrifuged at 14,000× *g* for 15 min at 4 °C. Imidazole (20 mM final concentration) was added to the supernatants before loading on a previously equilibrated 1 mL Ni-NTA agarose affinity resin (Chelating Sepharose FastFlow, GE Healthcare, Buc, France).

After 2-h rotation at room temperature (RT), the columns were washed with 25 mL of solubilization buffer (50 mM Tris-HCl pH 8, 100 mM NaCl, 20 mM imidazole, and containing 8 M urea only for recombinant SEA purified from inclusion bodies). Elution of the His-tagged recombinant proteins was performed with 5 × 2 mL of elution buffer (50 mM Tris-HCl pH 8, 100 mM NaCl, 500 mM imidazole, containing 8 M urea only for recombinant SEA from inclusion bodies).

The eluted recombinant SEA, SEG, SEH, and SEI fractions were dialyzed twice in 5 L of 50 mM potassium phosphate buffer pH 7.4 (and containing 0.2 M arginine for renaturation, only for recombinant SEA toxin purified from inclusion bodies) once at RT for 2 h, then at 4 °C for 15 h.

The protein purities were assessed by SDS PAGE (Phast system, GE Healthcare, Buc, France) and/or gel electrophoresis (Agilent protein 230 kit with Agilent 2100 Bioanalyzer, Agilent Technologies Inc., Santa Clara, CA, USA). Protein concentrations were assessed both by using the bicinchoninic acid protein assay (BCA protein assay kit, Thermo Fisher Scientific, Illkirch, France) and by measuring absorbance at 280 nm (recombinant His-tagged SEA: theoretical molecular extinction coefficient ε_theo_ = 37,945 M^−1^.cm^−1^ and theoretical molecular weight MW_theo_ = 28,290 Da; recombinant His-tagged SEG: ε_theo_ = 30,495 M^−1^.cm^−1^ and MW_theo_ = 28,237 Da; recombinant His-tagged SEH: ε_theo_ = 26,485 M^−1^.cm^−1^ and MW_theo_ = 26,313 Da; recombinant His-tagged SEI: ε_theo_ = 34,840 M^−1^.cm^−1^ and MW_theo_ = 26,116 Da).

The recombinant SEA, SEG, SEH, and SEI toxins (obtained from *E. coli* cytosol) were also characterized using mass spectrometry by bottom–up proteomics analysis (trypsin digestion) (D. Lefebvre et al., in revision, Journal of Agricultural and Food Chemistry).

### 5.3. Production of Monoclonal Antibodies against SEA, SEG, SEH, and SEI

All experiments were performed in compliance with French and European regulations on care and protection of laboratory animals (European Community (EC) Directive 2010/63/UE, French law 2001–486, 6 March 2018) and with the agreements of the Ethics Committee of the Commissariat à l’Energie Atomique (CEtEA “Comité d’Ethique en Expérimentation Animale” n° 44) no.12-026 and 15–046 delivered to S. Simon by the French Veterinary Services and CEA agreement D-91-272-106 from the Veterinary Inspection Department of Essonne (France).

Four groups of four *Biozzi* mice were immunized 4 times at 3-week intervals with respectively 1 μg of recombinant SEA toxin in alum adjuvant (subcutaneous injection) or with 10 μg of recombinant SEG, SEH, or SEI toxin in alum adjuvant (intraperitoneal (i.p.) injection). Immunization with non-inactivated toxins was chosen to raise better antibodies directed against native toxin (formalin or heat inactivation can denature, modify, or hide epitopes) and literature data showed that such immunizations with non-inactivated toxins were possible and induced immune response without reported toxicity [36,37,38,56,57,58,59,60]. Mice were bled before the first immunizations and two weeks after each immunization. The polyclonal anti-SEA, SEG, SEH, and SEI responses were evaluated using specific enzyme immunoassays (see below and Appendix A). The mice showing the best immune response were selected for the preparation of monoclonal antibodies (mAbs) and given a daily intravenous (i.v.) booster injection of SE toxins for three days (1 μg i.v. boost per mouse and per day for SEA; 50 μg i.p. boost per mouse and per day for SEG, SEH, and SEI). Two days after the last boost, hybridomas were produced by fusing spleen cells with NS1 myeloma cells, as previously described [61]. Hybridoma culture supernatants were screened for antibody production by enzyme immunoassay (see below and Appendix A). Selected hybridomas were subsequently cloned by limiting dilution. MAbs were produced from hybridoma culture supernatants and further purified by protein A or protein G affinity chromatography using the AKTAxpress system (GE Healthcare, Buc, France). The types of heavy and light chains of the antibodies obtained were determined by ELISA (Pierce rapid ELISA mouse mAb isotyping kit, Thermo Fisher Scientific, Illkirch, France) following the instructions provided in the kit. The purities of mAbs were assessed by SDS-PAGE in reducing and non-reducing conditions (Phast system, GE Healthcare, Buc, France) and/or gel electrophoresis (Agilent protein 230 kit with Agilent 2100 Bioanalyzer, Agilent Technologies Inc., Santa Clara, CA, USA) (Appendix A).

### 5.4. Labeling of Proteins with Biotin

Monoclonal antibodies, recombinant SE toxins, and commercial SEA (AT101red, Toxin Technology, Sarasota, FL, USA) were labeled with biotin for use as conjugates in enzyme immunoassays. Antibodies were incubated for 1 h at RT in 0.1 M borate buffer pH 8.5 with a 20-molar excess of biotin-N-hydroxysuccinimide ester (biotin-NHS, Sigma-Aldrich, Saint-Louis, MO, USA) dissolved in anhydrous *N,N*-dimethylformamide (DMF). A 10-molar excess of biotin-NHS was used for SEA, SEG, and SEI biotinylation and a 5-molar excess was used for SEH biotinylation. The reaction was stopped after 1 h at RT by the addition of 1 M Tris-HCl pH 8 for 10 min at RT. Finally, the conjugate was diluted in EIA buffer (0.1 M phosphate buffer pH 7.4 containing 0.15 M NaCl, 0.1% bovine serum albumin and 0.01% sodium azide) and stored at −20 °C until use.

### 5.5. Evaluation of Polyclonal Response and Screening of mAbs in Hybridoma Supernatants

To titrate the polyclonal response in mouse sera and to screen hybridoma culture supernatants, 50 μL of 10-fold serial dilutions of sera or hybridoma culture supernatants in EIA buffer were transferred into microtiter plates (Nunc MaxiSorp™, Thermo Fisher Scientific, Illkirch, France) previously coated with goat anti-mouse Ig(G+M) antibodies (Jackson Immunoresearch, Ely, UK). Then, 50 μL of biotinylated toxins in EIA buffer (200 ng/mL for SEA, 200 ng/mL for SEI, 100 ng/mL for SEH, and/or 40 ng/mL for SEG) was added before overnight reaction at 4 °C. After 3 washes (washing buffer: 10 mM potassium phosphate buffer pH 7.4, 0.05% Tween20), plates were reacted for 2 h at RT with 100 μL per well of acetylcholinesterase (AChE)-labeled streptavidin conjugate (home-made, 1 Ellman unit (EU)/mL [62]). After 1-h incubation at RT followed by five washing cycles, 200 μL of Ellman’s reagent [63] was added, and the absorbance was measured at 414 nm after 30 min.

To choose specific mAbs throughout the hybridoma selection and cloning process, a double screening immunotest (or triple for SEA) was performed. The first immunotest used the target SE as conjugate (for example, biotinylated recombinant SEG for anti-SEG mAb selection), and the second one used a mixture of the three other non-specific SEs (e.g., a mixture of biotinylated recombinant SEA, SEH, and SEI toxins for anti-SEG mAb selection). Exception was made for SEA mAb selection where a triple screening was performed, with biotinylated in-house recombinant SEA, biotinylated commercial SEA (Toxin Technology, Sarasota, FL, USA), and a mixture of biotinylated recombinant SEG, SEH, and SEI toxins.

### 5.6. Determination of mAb Affinity

The affinity of monoclonal antibodies for their target SE toxin was determined by means of bio-layer interferometry (BLI) using the Octet RED96e system from FortéBio (Sartorius, Aubagne, France) in a quantitative multiple-concentration kinetic assay performed at 25 °C.

First, a series of 8 anti-mouse immunoglobulin (Fc specific) capture (AMC) biosensors (from FortéBio, Sartorius, Aubagne, France) were dipped into equilibration buffer (100 mM potassium phosphate buffer pH 7.4, containing 0.1% bovine serum albumin, 0.15 M NaCl, 0.02% Tween20, and 0.01% sodium azide) for at least 10 min.

Then, the assay consisted of 4 cycles of 6 steps, in a 96-well format, enabling mAb affinity measurements for 4 different antibodies in one experiment (i.e., each biosensor was used 4 times). (1) First, biosensors were pre-conditioned/regenerated by being alternatively dipped into pH 1.3 glycine solution and pH 7.4 equilibration buffer every 20 sec for a total of 120 sec to remove the loaded immunoglobulins (IgGs). (2) Second, biosensors were dipped into equilibration buffer for 300 s to be equilibrated. (3) The next step consisted in loading sensors with anti-SE murine monoclonal IgG at 3 μg/mL in equilibration buffer for 250 sec. (4) A baseline was measured before immobilizing the ligand by dipping the biosensors into equilibration buffer for 300 sec. (5) Then, the 8 parallel prepared sensors were dipped into wells containing the target commercial or lab-made recombinant SE toxin in equilibration buffer in a 2-fold titration from 10 nM to 0.156 nM (and 0 nM) for 900 s to measure association kinetics (k_on_). (6) Next, sensors were placed in equilibration buffer for a further 1000 s to measure dissociation (k_off_).

After data processing (including reference subtraction using the 0 nM trace), the association and dissociation traces were fitted with the software FortéBio Data Analysis Software version 10.0 (Sartorius, Aubagne, France) using a 1:1 binding model with the global fitting function (grouped by cycle). Values of k_on_ and k_off_ were extracted from the curve-fitting analysis. K_D_ values were calculated as k_off_/k_on_.

### 5.7. Sandwich Enzyme Immunoassays

To identify the best pairs of mAbs to be used in a two-site immunometric test, a combinatorial analysis was carried out using each mAb either immobilized on the solid phase for the capture or as biotin-labeled as the tracer, using commercial SEA (AT101red, Toxin Technology, Sarasota, FL, USA) and in-house recombinant SEA, SEG, SEH, or SEI proteins as targets. Then, 96-well microplates (Nunc MaxiSorp™, Thermo Fisher Scientific, Illkirch, France) were coated overnight at RT with 100 μL of each of the different mAbs at 10 μg/mL in 0.05 M potassium phosphate buffer pH 7.4. Then, the plates were saturated with EIA buffer and stored at 4 °C until use. Target toxin samples were diluted in EIA buffer at 2 ng/mL and 0.4 ng/mL. To screen for the nonspecific recognition of toxins by mAbs, a mixture of the 3 other SE toxins was tested for each evaluated mAb pair (diluted in EIA at a final concentration of 2 ng/mL for each toxin, i.e., final 6 ng/mL of SEs). Duplicates of 50 μL of each toxin dilution were transferred into the washed coated microtiter plates together with 50 μL of biotinylated anti-SE mAb (200 ng/mL, i.e., 100 ng/mL final). After reaction at 4 °C overnight, followed by 3 washing cycles, plates containing the biotinylated conjugates were reacted for 1 h at RT with 100 μL per well of 1 EU/mL of AChE-labeled streptavidin. After 5 washes, AChE activity was detected by Ellman’s colorimetric method at 414 nm after 30 min (for SEG and SEH) or 1 h (for SEA and SEI).

The optimized enzyme immunoassay is a 3-h sequential format used to quantify SE toxins in samples (*S. aureus* culture supernatants or food extracts). Plates were coated with anti-SEA, anti-SEG, anti-SEH, or anti-SEI mAbs as described previously. After 3 washing cycles, duplicates of 100 μL of serial dilutions of samples were transferred into microtiter plates and incubated for 1 h at room temperature. Three-fold serial dilutions of commercial SEA and SEB and lab-made recombinant SEA, SEG, SEH, or SEI standards were prepared in the same buffer as for samples (in LB medium, 0.22 μm filtered BHI medium or EIA buffer), and deposited (100 μL in duplicates) onto each immunoplate to enable the quantification of SE toxins in samples present on the same 96-well plate. Then, the plates were washed 3 times, and 100 μL of biotinylated anti-SE mAb (100 ng/mL final in EIA buffer) was added. After reaction for 1 h at RT, followed by 3 washing cycles, immunoplates were reacted for 30 min at RT with 100 μL per well of 1 EU/mL of AChE-labeled streptavidin in EIA buffer and stained as previously described. Alternatively, immunoplates could also be reacted for 30 min at RT with 100 μL per well of poly-horseradish peroxidase-labeled streptavidin (Thermo Fisher Scientific, Illkirch, France) diluted 1/50,000 in EIA buffer without sodium azide. After 5 washes, plates were incubated for 30 min at RT with substrate solution containing tetramethylbenzidine (1-step ultra TMB, Thermo Fisher Scientific, Illkirch, France) and, after the addition of 100 μL of 2 M sulfuric acid to each well, absorbance was read at 450 nm in a microplate reader (Bio-Tek, Winooski, VT, USA).

The same 3-h sequential format with poly-horseradish peroxidase-labeled streptavidin detection was applied using anti-SEB monoclonal antibodies previously produced in the laboratory [47] (SEB27 as capture antibody and SEB26 as biotinylated tracer antibody) to complete detection of SEs in *S. aureus* supernatants.

### 5.8. Theoretical Limits of Detection and Quantification

For all immunoassay formats, LoD and LoQ were calculated using GraphPad Prism software with a nonlinear regression model using a two-site binding saturation curve fit (total and nonspecific binding). LoD is defined as the lowest calculated toxin concentration giving a signal greater than nonspecific binding (mean of eight measurements of unspiked EIA buffer/matrix) + 3 standard deviations (99.9% confidence). LoQ is defined as the lowest calculated toxin concentration giving a signal greater than nonspecific binding (mean of eight measurements of unspiked EIA buffer/matrix) + 10 standard deviations (99.9% confidence).

### 5.9. Repeatability and Reproducibility Assays

The intra-assay (repeatability) coefficients of variation were determined by assaying six times, on the same day, with the same instruments and reagents, duplicates of five different SE concentrations in EIA buffer within the working range (quality control concentrations from 1× to 2× experimental LoQ to the beginning of binding saturation). Inter-assay (in-house reproducibility) coefficients were determined by repeating this experiment on five different days. Accuracy (bias) was determined by measuring, throughout this repeatability and reproducibility study, the difference between the expected and measured quality control concentrations, using standard titration curves for all experiments.

### 5.10. Food and Clinical Sample Preparation

The human sample (diarrhea, stool, vomit) collection was conducted in accordance with the Declaration of Helsinki and its later amendments, and signed informed consent was obtained from all volunteers (who are the authors of this article). Sample collection was non-invasive and was directly performed by the volunteers themselves.

Commercial semi-skimmed milk (Lactel), pure pressed apple juice (Jardin Bio), Morbier cheese (E. Leclerc), chili with rice without meat (refectory 2018), chili with beef without rice (Brand Repère Côté Table), and ham (Herta) were purchased. SED naturally contaminated cheese (Morbier 08BAC553) was obtained from the National Reference Laboratory for CPS (ANSES, France). Two freeze-dried Emmental cheese powders (named 55 and 469) came from the proficiency testing trial organized by the EURL for CPS in 2019. Three Certified Reference Materials (CRM): blank cheese (IRMM 359a), cheese containing 0.1 ng SEA/g (IRMM 359b) and cheese containing 0.25 ng SEA/g (IRMM 359c) (https://crm.jrc.ec.europa.eu) [42,43,44] were also used. Before use, CRM were reconstituted as described in the quality certificate, and the same protocol was applied for freeze-dried Emmental cheese powders.

Liquid samples were buffered by the addition of a one-tenth volume of 10X EIA buffer. Solid samples were prepared by grinding with glass balls (two cycles at 6000 movements per minute for 2 min) using an Ultra-Turrax Tube Drive (IKA, Staufen, Germany) to obtain a 10% (*W*/*V*) homogenate in distilled water. Homogenates were centrifuged for 10 min at 2500× *g*. Supernatants were collected and buffered by addition of a one-tenth volume of 10X EIA buffer.

All buffered matrices were either used directly (pure 10% *W*/*V* homogenate) or diluted 2- or 5-fold in EIA buffer.

Lab-made recombinant enterotoxins SEA, SEG, SEH, or SEI were directly spiked at different concentrations in these buffered (pure, 2-, and 5-fold diluted) matrices. Then, 100 μL was transferred into the coated microtiter plates and analyzed as previously described.

### 5.11. Immunochromatographic Assays

The test is based on one-step immunochromatography using mAb coupled to colloidal gold particles. Colloidal gold-labeled anti-SE antibodies were prepared by adding 100 μL of BioReady 40 nm Gold Bare (citrate) Nanospheres (NanoComposix, San Diego, CA, USA) to 50 μL of mAb at 500 μg/mL in 20 mM borate buffer pH 9.0. The reaction mixture was incubated for 1 h at 20 °C in the dark, leading to the ionic adsorption of antibodies onto the surface of the colloidal gold particles. Then, 850 μL of 2 mM potassium phosphate buffer pH 7.4 containing 0.1% casein and 0.01% sodium azide (phosphate casein buffer) was added, and the mixture was centrifuged at 15,000× *g* for 20 min at 4 °C. The supernatant was discarded, and the pellet was washed with 1 mL of phosphate casein buffer. After a second centrifugation step, the pellet was resuspended in 250 μL of phosphate casein buffer, sonicated for a few seconds, and stored at 4 °C in the dark.

The test strip (0.5 cm wide and 4.5 cm long) is composed of 3 parts, (i) a sample pad (Standard 14 from Whatman, GE Healthcare, Buc, France) (0.5 cm long), (ii) a nitrocellulose membrane (PRIMA 40, Whatman, GE Healthcare, Buc, France) (2.5 cm long), and (iii) an adsorption pad (cellulose-grade 470, Whatman, GE Healthcare, Buc, France) (1.5 cm long), all attached to a backing card. The detection zone contains immobilized goat anti-mouse IgG antibodies (Jackson Immunoresearch, Ely, UK) as the control line and anti-SE antibodies as the test line (SEA7, SEG41, SEH14, or SEI44 at 1 mg/mL solution in 10 mM sodium phosphate buffer pH 7.4 containing 0.15 M NaCl) dispensed at 1 μL/cm using an automatic dispenser (Biojet XYZ 3050, BioDot, Chichester, United Kingdom). Saturation, drying, pad assembling, and cutting of the strips were achieved as previously described [64]. The assay was performed at room temperature in a 96-well microtiter plate by mixing 100 μL/well of the toxin sample with 10 μL (for SEA12) or 5 μL (for SEG27, SEH19, and SEI32) of 100 μg/mL colloidal gold-labeled antibody (all dilutions made in immunochromatographic (ICT) assay buffer: 0.1 M Tris-HCl pH 8 containing 0.15 M NaCl, 0.1% BSA, 0.5% Tween 20, 0.01% sodium azide and 1% 3-[(3-cholamidopropyl)dimethylammonio]-1-propanesulfonate (CHAPS)). After 5-min reaction of the mixture with gentle shaking, the lower part of the strip (i.e., the sample pad) was inserted into the well. A positive result appears as two lines and a negative result appears as a single upper control line. The detection limit corresponds to the lowest toxin concentration, showing a positive result detected by the naked eye after 30 min.

Multiplex immunochromatographic tests were produced in a pre-industrial format (strip plus cassette) by NG-Biotech (Guipry, France) using SEA7, SEB27, SEG41, SEH14, and SEI44 as immobilized capture mAbs on test lines, goat anti-mouse IgG antibodies as the control line, and SEA12, SEB26, SEG27, SEH19, and SEI32 as colloidal gold-labeled conjugate mAbs lyophilized on a single release zone. The assay was performed by dispensing 100 μL onto the cassette. Results were read by the naked eye after 30 min.

For immunochromatographic assays with *S. aureus* BHI cultures, supernatants were filtered through 0.22 μm, incubated overnight with 5% (*V*/*V*) Ehrlich ascites fluid to avoid any nonspecific reaction caused by protein A, and buffered by addition of a one-tenth volume of 10X ICT buffer (i.e., 1X final ICT buffer in supernatants) before being tested. Complex matrices were also buffered by the addition of a one-tenth volume of 10X ICT buffer before spiking with lab-made recombinant SEs and testing.

## Figures and Tables

**Figure 1 toxins-13-00130-f001:**
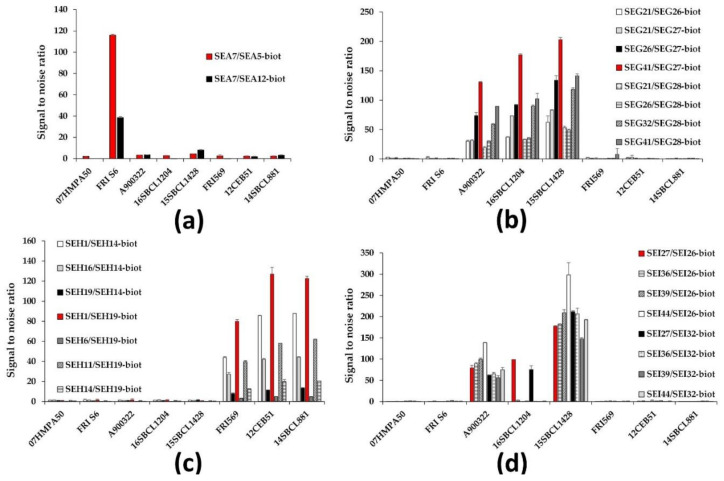
Comparison of the best selected immunoassays for the specific detection of staphylococcal enterotoxin type A, G, H or I (SEA, SEG, SEH, or SEI respectively) in *S. aureus* supernatants a 3-h sequential format. Eight *S. aureus* strains were grown at 37 °C for 16 h in Brain Heart Infusion (BHI) medium (for the selection of the best anti-SEA immunoassay in (**a**)) or Luria–Bertani (LB) medium (for anti-SEG (**b**), SEH (**c**), and SEI (**d**) immunoassay selection). The centrifuged and 0.22 μm-filtered supernatants from these *S. aureus* cultures (undiluted for SEA, SEG, and SEI immunoassays, 10-fold diluted in enzyme immuno-assay (EIA) buffer for SEH immunoassays) were detected with various mAb pairs in a 3-h sequential format with acetylcholinesterase (AChE)-labeled streptavidin and Ellman’s colorimetric method detection (see experimental procedures). Error bars represent standard deviations from a duplicate.

**Figure 2 toxins-13-00130-f002:**
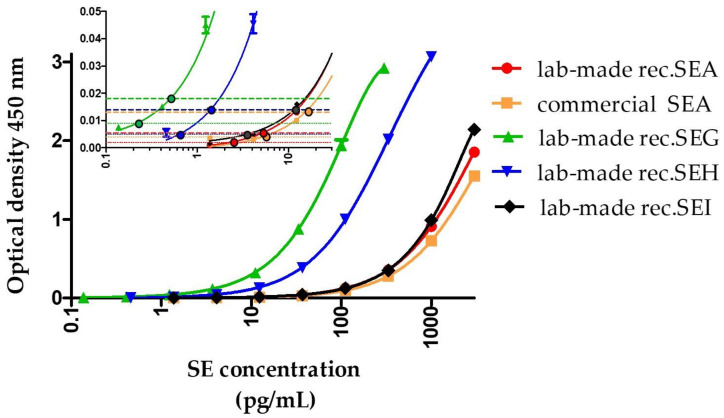
Representative standard curves obtained with each of the best sandwich enzyme immunoassays for the detection of SEA, SEG, SEH, or SEI using the 3-h sequential format. Dilutions of target toxins in EIA buffer were detected using the best-identified sandwich immunoassay (Table 3) using the 3-h sequential format with poly-horseradish peroxidase-labeled streptavidin detection. The insert shows the low concentration part of the curve. Colored spherical points indicate the theoretical limit of detection (LoD) and limit of quantification (LoQ) for each sandwich immunoassay. Colored thin dotted and dashed horizontal lines represent optical densities reached for the detection limit (signal greater than nonspecific binding + three standard deviations) and quantification limit (signal greater than nonspecific binding + 10 standard deviations), respectively. Error bars represent standard deviations from the duplicate.

**Figure 3 toxins-13-00130-f003:**
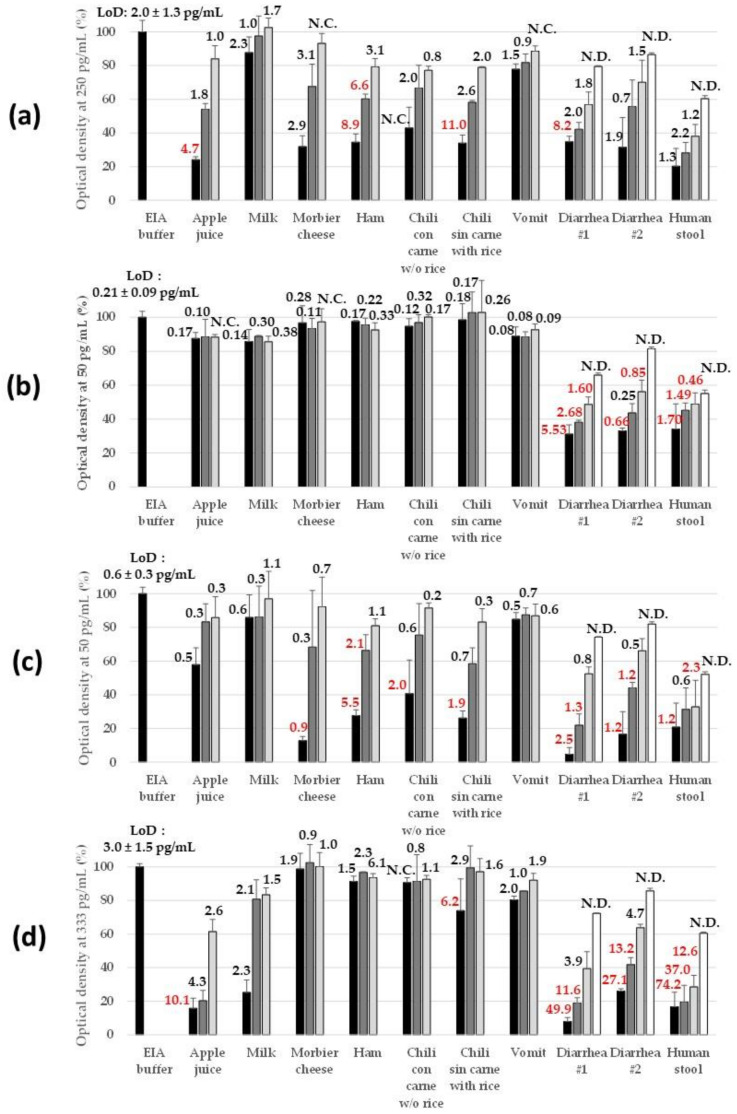
Robustness of the sandwich immunoassays in various complex matrices. Dilutions of target toxins, i.e., recombinant lab-made SEA at 250 pg/mL (**a**), SEG at 50 pg/mL (**b**), SEH at 50 pg/mL (**c**), and SEI at 333 pg/mL (**d**), were performed in EIA buffer (positive control reaching approximately 1 optical density (OD) signal), undiluted matrices (black bars), 2-fold diluted matrices in EIA buffer (dark gray bars), 5-fold diluted matrices in EIA buffer (clear gray bars), or 20-fold diluted matrices in EIA buffer (white bars). Undiluted liquid matrices consisted of raw matrices buffered by the addition of a one-tenth volume of 10X EIA buffer. Undiluted solid matrices consisted of 10% (*W*/*V*) ground homogenate prepared in water and buffered by the addition of a one-tenth volume of 10X EIA buffer. All these spiked samples were detected using the best-identified sandwich immunoassay (Table 3) using the 3-h sequential format with poly-horseradish peroxidase-labeled streptavidin detection. Results are expressed as a percentage of signal (optical density with subtraction of nonspecific binding) obtained when the target toxin is prepared in EIA buffer (black bar on the right side of each graph). Numbers located above the bars indicate the theoretical LoD (calculated as explained in methods) measured in the corresponding matrix and calculated from one experiment. Error bars represent standard deviations from two independent experiments. N.C., not calculable. N.D., not determined.

**Figure 4 toxins-13-00130-f004:**
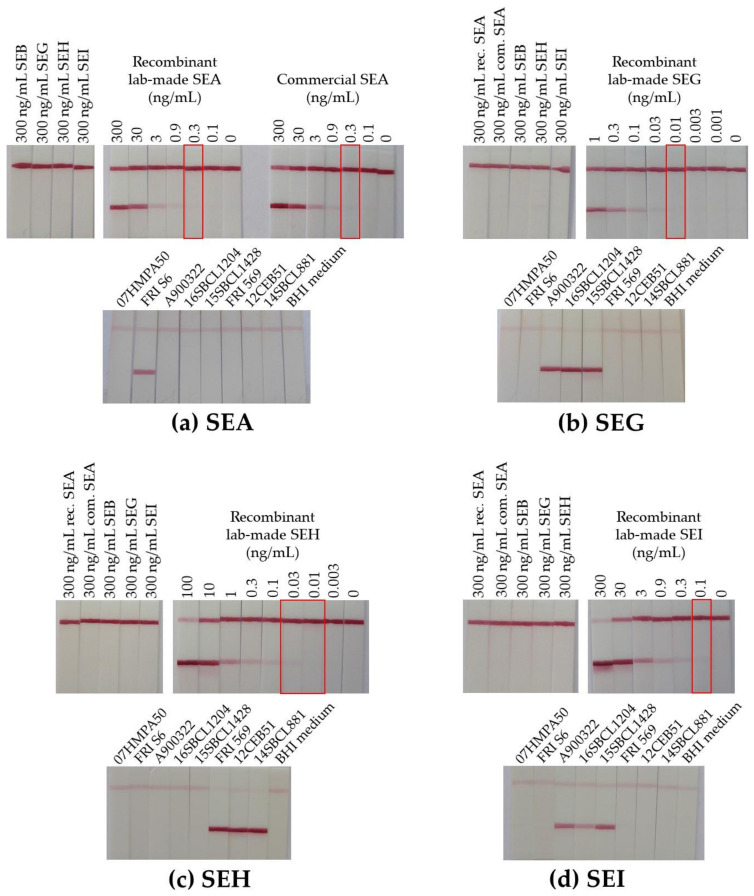
Specificity and sensitivity of detection of SEA, SEG, SEH and SEI using monoplex immunochromatographic tests after 30 min of migration. Dilutions of commercial SEA, commercial SEB, lab-made recombinant SEA, SEG, SEH, or SEI toxins were prepared in immunochromatographic (ICT) buffer before detection with the lab-made strips developed with the following capture/colloidal gold conjugate mAb pairs: SEA7/SEA12 in (**a**), SEG41/SEG27 (**b**), SEH14/SEH19 (**c**), and SEI44/SEI32 (**d**), respectively. BHI-cultured *S. aureus* supernatants were processed as described in the methods before performing the test. Overnight pre-incubation with immunoglobulin (IgG)-enriched Ehrlich ascites neutralizes the interfering protein A that is naturally produced by *S. aureus* and restores the right signal for SE toxin detection on the test line (with reduction of the control line intensity).

**Figure 5 toxins-13-00130-f005:**
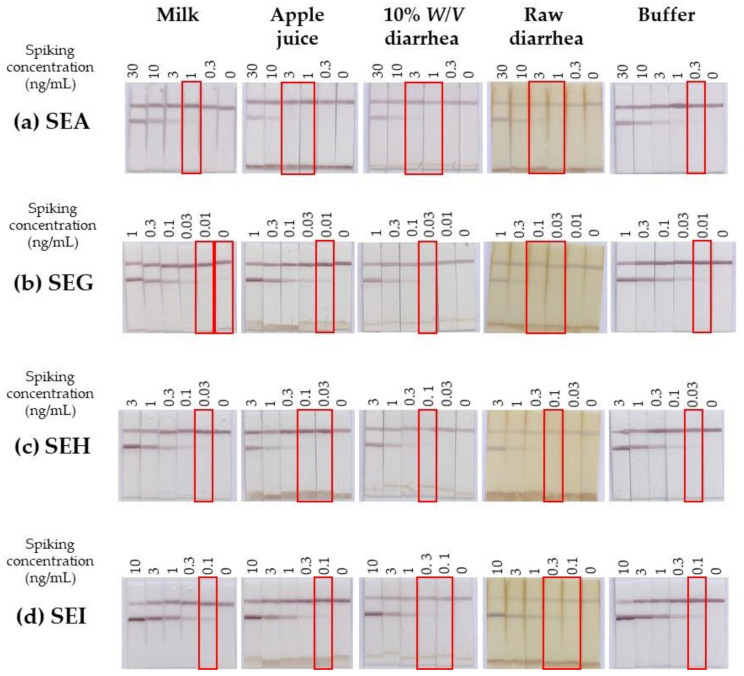
Robustness of the monoplex immunochromatographic assays in different complex matrices. Dilutions of lab-made recombinant SEA, SEG, SEH, or SEI toxins were prepared in immunochromatographic (ICT) buffer or in different matrices (semi-skimmed milk, apple juice, 10% *W*/*V* diarrhea homogenate, or raw diarrhea). These matrices were previously buffered by addition of a one-tenth volume of 10X ICT buffer. Then, samples were detected with the lab-made strips developed with the following capture/colloidal gold conjugate mAb pairs: SEA7/SEA12 in (**a**), SEG41/SEG27 (**b**), SEH14/SEH19 (**c**), and SEI44/SEI32 (**d**), respectively.

**Figure 6 toxins-13-00130-f006:**
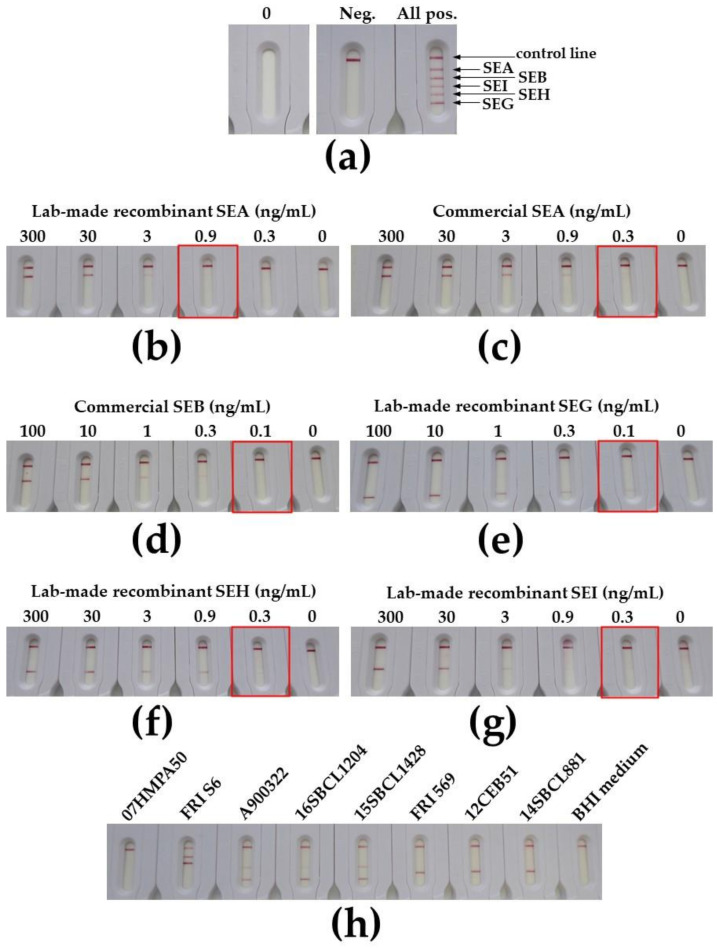
Detection of SEA, SEB, SEG, SEH, and SEI using multiplex immunochromatographic test after 30 min of migration. (**a**) Description of the manufactured multiplex cassette: 0, cassette before use; Neg., negative result; All pos., positive result for all enterotoxins (solution containing 3 ng/mL for each SEA, SEH, and SEI and 1 ng/mL for each SEB and SEG in ICT buffer). Serial dilutions with lab-made recombinant SEA in (**b**), commercial SEA (**c**), commercial SEB (**d**), lab-made recombinant SEG (**e**), SEH (**f**), or SEI (**g**) toxin were prepared in immunochromatographic (ICT) buffer before loading 100 μL onto the cassette. (**h**) BHI-cultured *S. aureus* supernatants were processed as described in the methods section before performing the test by loading 100 μL onto the cassette. The monoclonal antibody (mAb) pairs used in these manufactured (NG-Biotech, France) multiplex cassettes were the same as for the lab-made monoplex strips.

**Table 1 toxins-13-00130-t001:** List of *S. aureus* strains used in this study. NRL, National Reference Center. CPS, coagulase-positive staphylococci.

*S. aureus* Strain	SE Genes	Origin and Source
07HMPA50	Nontoxic coagulase-positive *S. aureus*	Commercial Camembert 1998 (NRL for CPS, ANSES, France)
FRI S6	*sea, seb*	Reference strain Food Research Institute, Frozen shrimp (M. S. Bergdoll [49])
A900322	*seg, sei, sep*	Reference strain (Clinical strain from a patient with TSS, NRC for staphylococcus, France)
16SBCL1204	*seg, sei*	Food poisoning outbreak, rice chicken (NRL for CPS, ANSES, France)
15SBCL1428	*seg, sei*	Food poisoning outbreak, cheese (NRL for CPS, ANSES, France)
FRI 569	*seh*	Reference strain Food Research Institute Nasal swab (Y.-C. Su, A.C.L. Wong [31])
12CEB51	*seh*	Food poisoning outbreak, Pot au feu (NRL for CPS, ANSES, France)
14SBCL881	*seh*	Food poisoning outbreak, mixed salad (NRL for CPS, ANSES, France)

**Table 2 toxins-13-00130-t002:** Comparison of theoretical LoD and LoQ of the best selected immunoassays presented in Figure 1. Dilution series of commercial SEA and target lab-made recombinant toxins were made in BHI medium for SEA, LB medium for SEG and SEI, or EIA buffer for SEH and were detected as described in experimental procedures using AChE-labeled streptavidin and Ellman’s colorimetric method. Theoretical LoD and LoQ were calculated (see Methods) from each experiment. LoD, limit of detection. LoQ, limit of quantification. N.C., not calculable.

Target Toxin	mAb Pair	Theoretical LoD (pg/mL)	Theoretical LoQ (pg/mL)
Lab-made rec. SEA	SEA7/SEA5-biot	2	10
Commercial SEA		3	14
Lab-made rec. SEA	SEA7/SEA12-biot	7	35
Commercial SEA		24	100
Lab-made rec. SEG	SEG21/SEG26-biot	6	23
	SEG21/SEG27-biot	5	16
	SEG26/SEG27-biot	N.C.	N.C.
	SEG41/SEG27-biot	8	24
	SEG21/SEG28-biot	30	105
	SEG26/SEG28-biot	3	9
	SEG32/SEG28-biot	4	9
	SEG41/SEG28-biot	4	10
Lab-made rec. SEH	SEH1/SEH14-biot	3	11
	SEH16/SEH14-biot	13	50
	SEH19/SEH14-biot	47	135
	SEH1/SEH19-biot	2	11
	SEH6/SEH19-biot	148	346
	SEH11/SEH19-biot	5	22
	SEH14/SEH19-biot	53	179
Lab-made rec. SEI	SEI27/SEI26-biot	63	163
	SEI36/SEI26-biot	59	163
	SEI39/SEI26-biot	34	103
	SEI44/SEI26-biot	58	160
	SEI27/SEI32-biot	57	184
	SEI36/SEI32-biot	41	122
	SEI39/SEI32-biot	41	131
	SEI44/SEI32-biot	30	95

**Table 3 toxins-13-00130-t003:** Sensitivity in buffer of the different sandwich enzyme immunoassays for the detection of SEA, SEB, SEG, SEH, or SEI in a 3-h sequential format. Dilutions of target toxins in EIA buffer were detected using the best-identified sandwich immunoassay using the 3-h sequential format with poly-horseradish peroxidase-labeled streptavidin detection. Theoretical LoD and LoQ were calculated as explained in methods. Experimental LoD and LoQ were defined as the lowest measured toxin concentration giving a signal greater than the nonspecific binding (NSB) + three standard deviations (SD) (with almost a 95% confidence) and NSB + 10 SD, respectively. Data represent the mean of n experiments performed independently.

			Capture/Tracer mAb Pair				
			SEA7/SEA5-biot		SEG41/SEG27-biot	SEH1/SEH19-biot	SEI27/SEI26-biot
			Lab-Made Recombinant SEA	Commercial SEA	Lab-Made Recombinant SEG	Lab-Made Recombinant SEH	Lab-Made Recombinant SEI
**Calculated estimation using fitting curves**	Theoretical LoD	(pg/mL)	2.0 ± 1.3 (*n* = 20)	5.4 ± 0.7 (*n* = 5)	0.21 ± 0.09 (*n* = 12)	0.59 ± 0.26 (*n* = 12)	3.0 ± 1.5 (*n* = 12)
		(fM)	70.7 ± 46.0 (*n* = 20)	199.4 ± 25.8 (*n* = 5)	7.4 ± 3.2 (*n* = 12)	22.4 ± 9.9 (*n* = 12)	114.9 ± 57.4 (*n* = 12)
	Theoretical LoQ	(pg/mL)	6.0 ± 3.0 (*n* = 20)	17.3 ± 6.1 (*n* = 5)	0.66 ± 0.16 (*n* = 12)	1.73 ± 0.72 (*n* = 12)	10.1 ± 5.1 (*n* = 12)
		(fM)	212.1 ± 106.0 (*n* = 20)	638.8 ± 225.3 (*n* = 5)	23.4 ± 5.7 (*n* = 12)	65.7 ± 27.4 (*n* = 12)	386.7 ± 195.3 (*n* = 12)
**Experimental evaluation**	Experimental LoD	(pg/mL)	4.1 (100%, *n* = 28)	12.3 (100%, *n* = 5)	0.41 (100%, *n* = 17)	1.37 (100%, *n* = 17)	4.1 (94.1%, *n* = 17)
		(fM)	144.9 (100%, *n* = 28)	454.2 (100%, *n* = 5)	14.5 (100%, *n* = 17)	52.1 (100%, *n* = 17)	157.0 (94.1%, *n* = 17)
	Experimental LoQ	(pg/mL)	12.3 (100%, *n* = 28)	37.0 (100%, *n* = 5)	1.23 (100%, *n* = 17)	4.12 (94.1%, *n* = 17)	37.0 (100%, *n* = 17)
		(fM)	434.8 (100%, *n* = 28)	1366.3 (100%, *n* = 5)	43.6 (100%, *n* = 17)	156.6 (94.1%, *n* = 17)	1416.8 (100%, *n* = 17)

**Table 4 toxins-13-00130-t004:** Analysis of accuracy, in-house reproducibility, and repeatability of the developed sandwich enzyme immunoassays using lab-made recombinant toxins and commercial SEA. SEA, SEG, SEH, and SEI toxins were respectively quantified using SEA7/SEA5-biot, SEG41/SEG27-biot, SEH1/SEH19-biot, and SEI27/SEI26-biot sandwich 3-h sequential format immunoassays with poly-horseradish peroxidase-labeled streptavidin detection. Accuracy (bias), repeatability, and in-house reproducibility were measured as described in methods. CV, coefficient of variation.

Target Toxin	Quality Control Standard Concentration	Accuracy (bias)	Repeatability (CV)	In-House Reproducibility (CV)
**Commercial SEA**	25 pg/mL (0.92 pM)	101.3%	14.1%	15.9%
	50 pg/mL (1.85 pM)	97.0%	6.1%	13.8%
	100 pg/mL (3.69 pM)	93.3%	6.4%	7.2%
	300 pg/mL (11.08 pM)	97.1%	4.6%	4.7%
	1000 pg/mL (36.93 pM)	96.7%	5.2%	5.5%
**Lab-made recombinant SEA**	25 pg/mL (0.88 pM)	95.0%	8.2%	9.5%
	50 pg/mL (1.77 pM)	95.5%	7.0%	8.0%
	100 pg/mL (3.53 pM)	91.5%	6.8%	6.7%
	300 pg/mL (10.60 pM)	92.1%	7.9%	7.9%
	1000 pg/mL (35.35 pM)	91.3%	9.6%	8.9%
**Lab-made recombinant SEG**	3 pg/mL (0.11 pM)	90.7%	6.9%	8.2%
	5 pg/mL (0.18 pM)	91.6%	8.3%	9.4%
	10 pg/mL (0.35 pM)	90.5%	6.5%	8.3%
	30 pg/mL (1.06 pM)	93.0%	6.5%	8.9%
	60 pg/mL (2.12 pM)	94.7%	6.9%	9.8%
**Lab-made recombinant SEH**	5 pg/mL (0.19 pM)	87.1%	9.1%	10.6%
	10 pg/mL (0.38 pM)	89.0%	7.1%	12.6%
	30 pg/mL (1.14 pM)	88.0%	6.1%	7.6%
	100 pg/mL (3.80 pM)	89.9%	5.4%	6.3%
	300 pg/mL (11.4 pM)	88.3%	7.0%	7.8%
**Lab-made recombinant SEI**	30 pg/mL (1.15 pM)	88.8%	8.6%	8.8%
	50 pg/mL (1.91 pM)	90.8%	6.4%	7.8%
	100 pg/mL (3.83 pM)	89.9%	5.1%	6.8%
	300 pg/mL (11.49 pM)	88.9%	5.1%	6.1%
	1000 pg/mL (38.29 pM)	88.3%	3.7%	5.3%

**Table 5 toxins-13-00130-t005:** Measured staphylococcal enterotoxins (SE) concentrations in culture supernatants from eight *S. aureus* strains grown in BHI medium. The supernatants were collected and processed as described in methods before quantification using the 3-h sequential format sandwich immunoassay with poly-horseradish peroxidase-labeled streptavidin detection. Concentrations were calculated with GraphPad Prism software using a nonlinear regression model (two-site binding saturation curve fit). Means and standard deviations for SE concentrations in supernatants were calculated from two experiments performed with two independent *S. aureus* cultures.

mAb pair	Target SE	Measured SE Concentrations (ng/mL eq. Recombinant SE) in Culture Supernatant from Strain:							
		07HMPA50	FRI S6	A900322	16SBCL1204	15SBCL1428	FRI 569	12CEB51	14SBCL881
SEA7/SEA5-biot	SEA	<LoD	0.18 ± 0.02	<LoD	LoD	<LoD	<LoD	<LoD	<LoD
SEB27/SEB26-biot	SEB	<LoD	13400 ± 2263	<LoD	<LoD	<LoD	<LoD	<LoD	<LoD
SEG41/SEG27-biot	SEG	<LoD	<LoD	2.6 ± 0.9	5.3 ± 1.6	3.3 ± 0.4	<LoD	<LoD	<LoD
SEH1/SEH19-biot	SEH	<LoD	<LoD	<LoD	<LoD	<LoD	103.6 ± 41.2	43.9 ± 14.4	82.7 ± 33.5
SEI27/SEI126-biot	SEI	<LoD	<LoD	8.1 ± 7.7	1.9 ± 0.6	5.7 ± 0.1	<LoD	<LoD	<LoD

**Table 6 toxins-13-00130-t006:** Performance of the immunoassays with real samples. LoD, limit of detection.

Context	Sample	EN ISO 19020		Enzyme Immunoassays				Expected Results
		(Qualitative Detection)						
		Vidas SETII (TV^(a)^)	Ridascreen SET Total (AU^(b)^)	SEA (pg/g)	SEG (pg/g)	SEH (pg/g)	SEI (pg/g)	
Certified reference materials [42,43,44]	IRMM-359a cheese	not detected	not detected	<LoD ^(c)^	<LoD ^(c)^	<LoD ^(c)^	<LoD ^(c)^	SEA, SEG, SEH, and SEI not detected
								Vidas SET II: SEA to SEE not detected
				<LoD ^(d)^	<LoD ^(d)^	<LoD ^(d)^	<LoD ^(d)^	Ridascreen ST: SEA to SEE not detected
	IRMM-359b cheese	1.35 TV (positive)	0.96 AU (positive)	38.4 ^(c)^	<LoD ^(c)^	<LoD^(c)^	<LoD ^(c)^	42 pg SEA/g (interval: 29–59 pg/g)
								Vidas SET II: 1.14 TV (0.47–1.53 TV)
				41.8 ^(d)^	<LoD ^(d)^	<LoD^(d)^	<LoD ^(d)^	Ridascreen ST: 0.61 AU (0.28– 1.11 AU)
	IRMM-359c cheese	2.08 TV (positive)	2.03 AU (positive)	115.4 ^(c)^	<LoD ^(c)^	<LoD ^(c)^	<LoD ^(c)^	102 pg SEA/g (interval: 81–145 pg/g)
								Vidas SET II: 1.97 TV (1.10–2.42 TV)
				110.5 ^(d)^	<LoD^(d)^	<LoD ^(d)^	<LoD ^(d)^	Ridascreen ST: 1.36 AU (0.45–2.31 AU)
Samples from French CPS NRL collection	Emmental cheese 55	1.45 TV (positive)	0.92 AU (positive)	123.3 ^(c)^	<LoD ^(c)^	<LoD ^(c)^	<LoD ^(c)^	80 pg SEA/g (spiking concentration)
	Emmental Cheese 469	1.45 TV (positive)	0.92 AU (positive)	185.4 ^(c)^	<LoD ^(c)^	<LoD ^(c)^	<LoD ^(c)^	80 pg SEA/g (spiking concentration)
	Morbier cheese 08BAC553	1.06 TV (positive)	0.66 AU (positive)	<LoD ^(c)^	<LoD ^(c)^	<LoD ^(c)^	<LoD ^(c)^	Natural contamination with 180 pg SED/g

^(a)^ TV: test Value, TV ≥ 0.13 means SEA to SEE detected. ^(b)^ AU: absorbance unit. Cut-off = AU of negative control + 0.15 AU (calculated for each experiment). AU ≥ cut-off means SEA to SEE detected. ^(c)^ Tested in our laboratory (CEA, France). ^(d)^ Tested at CPS NRL laboratory (ANSES, France) after transfer of the method.

## Data Availability

Data is contained within the article or Appendix A. The data presented in this study are available in Féraudet Tarisse, C.; Goulard-Huet, C.; Nia, Y.; Devilliers, K.; Marc, D.; Dambrune, C.; Lefebvre, D.; Hennekinne, J.-A.; Simon, S. Highly Sensitive and Specific Detection of Staphylococcal Enterotoxins SEA, SEG, SEH and SEI by Immunoassay. *Toxins*
**2021**, *13*, 130. https://doi.org/10.3390/toxins13020130.

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
