# Peer review of "Highly Sensitive and Specific Detection of Staphylococcal Enterotoxins SEA, SEG, SEH, and SEI by Immunoassay"

_toxins, 2021, doi:10.3390/toxins13020130_

Round 1

Reviewer 1 Report

Manuscript number: toxins-1053840

Highly sensitive and specific detection of staphylococcal enterotoxins SEA, SEG, SEH and SEI by immunoassay

Authors developed multi-chromatographic lateral immunoassay for staphylococcal food poisoning in food. The manuscript includes a lot of data and they were accomplished with logical design of experiments, but it did not show scientifically-valid evidences for the conclusions appropriately. The manuscript includes routine analysis method and results of ELISA for SEA, SEG, SHE and SEI. The new organization and explanations are necessary. It needs major revisions.

  1. The advantage and needs of this study is not clear to the readers.
  2. The characteristics and physical chemical data for antibodies should be included and explained.
  3. In Table 4, Why there are almost negative bias for accuracy? Could you explain it?
  4. Please discuss about the different kd values for each SEs and related hypothesis using references.
  5. Table captions are too long. The experimental methods are duplicate in Methods section.
  6. How could you explain about the big difference of LoD between theoretical and experimental assay for SEA ?
  7. Some validation data are missing. Stability, Cross reactivity and linearity of calibration curves.

Reviewer 2 Report

The authors of " Highly sensitive and specific detection of staphylococcal enterotoxins SEA, SEG, SEH and SEI by immunoassay" developed an highly sensitive ELISA test for the detection of SEs in contaminated food and human samples and described, for the first time, an immunochromatographic multiplex assay for the same toxins.

The paper is well written, the study design is well structured, the scientific relevance is high due, in primis,  to the diagnostic application of ELISA, greater efforts should be made in a future paper to fine-tune system on biological samples.

Minor revisions are:

Line 52: "Low CPS levels cannot exclude the presence of preformed SEs in food..." add a reference.

Figure 3: Captions format are too small and barely legible.

Best Regards

Round 2

Reviewer 1 Report

The authors tried to revise the manuscript according to my suggestions. The manuscript could be accepted now.

Author Response

We thank reviewer 1 for taking time to review the revised manuscript and for his/her positive evaluation.